# Transcriptome Analysis Reveals Mycelial and Fruiting Responses to Lithium Chloride in *Coprinopsis cinerea*

**DOI:** 10.3390/jof10020140

**Published:** 2024-02-09

**Authors:** Po-Lam Chan, Hoi-Shan Kwan, Yichun Xie, Ka-Hing Wong, Jinhui Chang

**Affiliations:** 1Research Institute for Future Food, The Hong Kong Polytechnic University, Hong Kong SAR, China; po-lam-kathy.chan@polyu.edu.hk; 2Food Research Centre, School of Life Sciences, The Chinese University of Hong Kong, Shatin, New Territories, Hong Kong SAR, China; hoishankwan@cuhk.edu.hk (H.-S.K.);; 3State Key Laboratory of Agrobiotechnology, The Chinese University of Hong Kong, Shatin, New Territories, Hong Kong SAR, China; 4Department of Food Science and Nutrition, The Hong Kong Polytechnic University, Hong Kong SAR, China

**Keywords:** glycogen synthase kinase3, fruiting body development, environmental signals, kinases, RNA-Seq

## Abstract

Lithium chloride (LiCl) has been used in signalling and molecular studies of animals, plants, and yeast. However, information on its roles in basidiomycetous fungi is still limited. In this study, we used RNA-Seq to study the effects of LiCl on *Coprinopsis cinerea*. LiCl enhanced mycelial growth and inhibited fruiting body formation in *C. cinerea*. RNA-Seq of the LiCl-treated *C. cinerea* resulted in a total of 14,128 genes. There were 1199 differentially expressed genes (DEGs) between the LiCl-treated samples and control samples in the mycelium stage (the first time point), and 1391 DEGs were detected when the control samples were forming hyphal knots while the treated samples were still in the mycelium (the second time point). Pathway enrichment analysis of the DEGs revealed a significant association between enhanced mycelium growth in the LiCl-treated *C. cinerea* and metabolic pathways. In addition, the DEGs involved in cellular process pathways, including “cell cycle-yeast” and “meiosis-yeast”, were identified in suppressed *C. cinerea* fruiting body formation by LiCl under favourable environmental conditions. As LiCl can predominantly inhibit the activity of glycogen synthase kinase3 (GSK3), our findings suggest that LiCl affects the expression of genes involved in fruiting body initiation and cellular processes by inhibiting GSK3 activity which is essential for fruiting body formation.

## 1. Introduction

*Coprinopsis cinerea* (Schaeff.) Redhead, Vilgalys & Moncalvo is a model fungus for studying the development and growth regulation in homobasidiomycetous fungi. *C. cinerea* completes its life cycle in two weeks under laboratory conditions. There are at least six developmental stages in the life cycle of *C. cinerea*: mycelium, initial, stage 1 primordium, stage 2 primordium, young fruiting body, and mature fruiting body with spore formation. The gill, and frequently the entire cap, of *C. cinerea* undergoes autolysis to generate an inky black fluid at maturity [1]. Due to its rapid morphogenesis, *C. cinerea* is a favourable model for genetic studies of fruiting body development [2]. In particular, #326 (*AmutBmut pab1-1*) is a homokaryotic strain that forms a fruiting body without mating, making it especially suitable for molecular studies of basidiomycetous fungi under different environmental conditions [3].

The development of *C. cinerea* depends on the sensing of environmental conditions, including light, temperature, humidity, and nutrition [1,4]. The natural substrate of *C. cinerea* is horse dung. After the mycelium reaches the surface of horse manure, the habitat changes drastically. Circadian rhythms, especially light, not only trigger adaptation to this changing environment, but also regulate sexual development. The transformation from the mycelium to hyphal knot requires light, glucose depletion, and a lower temperature [1]. Blue light is required initiate body fruiting, primordial maturation, and karyogamy, and the fruiting body ensures the eating and thus spreading of spores by animals in the early morning [1,5]. The question of how environmental signals stimulate regulators of adaptation and reproduction has attracted broad research interest. Kinases and transcription factors are among the most intensively studied regulators. However, the functions of kinases, which play a central role in signalling and metabolic pathways, have not been described on a genome-wide scale of *C. cinerea* [4,6].

Lithium chloride (LiCl) is a non-toxic, water-soluble alkali metal salt [7]. Although non-essential to most organisms, it has a variety of functions to plants, bacteria, and fungi. Kado and Heskett [8] used LiCl to isolate plant pathogenic bacteria from soil and stimulate plant growth. LiCl was used to study lithium bioaccumulation in *Pleurotus ostreatus* mycelial growth for the development of biotechnological products. In addition, LiCl is a promising compound to enrich the lithium content in mushrooms and affect fungal growth [9]. Apart from mushroom-forming fungi, LiCl has also been reported to inhibit the growth of *Trichoderma* species; however, several wood-decaying basidiomycetes species can tolerate LiCl at 6 g/L [10].

LiCl was used to study the relationship between the nutrient source and signalling and metabolic pathways that affect hyphal outgrowth in *Candida albicans* and *Saccharomyces cerevisiae* [11,12]. Our previous study has reported that a particular concentration of LiCl at 3 g/L inhibits fruiting body formation in *C. cinerea* [13]. However, the mechanism underlying the effect of LiCl in fruiting body development remains unknown. LiCl has been used to study the characteristics of glycogen synthase kinase (GSK3) in humans and animal models [14,15]. The inhibition of GSK3 is the predominant action of LiCl [16]. In this study, we investigated the impact of LiCl promotion on mycelial growth and fruiting body initiation in *C. cinerea*. We used RNA-Seq to examine the effects of LiCl on the transcriptional responses of *C. cinerea* in mycelial growth, and fruiting body formation. We propose that LiCl affects the phosphorylation of GSK3, which in turn affects fruiting body development in *C. cinerea*, and we suggest that GSK3 activity is essential for fruiting body formation by affecting the expression of genes involved in fruiting body induction and cellular processes.

## 2. Materials and Methods

### 2.1. Strain and Cultivation Conditions

*C. cinerea* strain #326 (A43mut B43mut pab1-1), a homokaryotic fruiting strain, was grown on yeast extract–malt extract–glucose (YMG) medium solidified with 1.5% (*w*/*v*) agar [17] at 37 °C for 5–6 days until it reached the edge of plates. In the treatment experiments, a piece of mycelium 5 mm in diameter was cut from the stock plates and inoculated on plates spread with different concentrations (8.5 mM, 17 mM, 35 mM, 70 mM, 140 mM) of LiCl (99%, Sigma-Aldrich, St. Louis, MO, USA). To prepare different concentrations of LiCl, LiCl was dissolved in autoclaved water after filter sterilisation and covered with cellophane. A concentration of potassium chloride (KCl) up to 70 mM was also prepared for comparison. Autoclaved water was added to the controls. The mycelium in the treatment experiments was incubated at 37 °C in total darkness for 4 days and then harvested for RNA extraction and transcriptome sequencing. Another set of RNA extraction was carried out to the control plates after the formation of hyphal knots. After the mycelium reached the edge of plates, the plates were incubated at 28 °C in a 12/12 h light/dark cycle for fruiting body development. Each treatment experiment was carried out in independent biological triplicates under the same experimental conditions.

### 2.2. RNA Isolation, Library Preparation, and Sequencing

For transcriptome sequencing, samples of *C. cinerea* were harvested at two time points. Samples were harvested in the mycelium stage after four days of incubation in total darkness, before mycelium reached the edge of plates, and also in the hyphal knot stage after 3–4 days of incubation in a 12/12 h light/dark cycle after the mycelium reached the edge of plates. Samples collected were frozen in liquid nitrogen immediately and stored at −80 °C before RNA extraction by using the RNeasy Plant Mini Kit (Qiagen, Hilden, Germany). RNA was first treated with the TURBO DNA-free kit according to the manufacturer’s instructions. The quality and quantity of the extracted RNA were assessed on 1.5% agarose gels and an Agilent 2100 bioanalyser with the Agilent RNA 6000 Nano Kit. RNA with an RNA integrity number (RIN) greater than 8.0 was used for transcriptome library construction. RNA sequencing libraries were constructed by using the TruSeq RNA Sample Prep Kit v2 (Illumina, San Giego, CA, USA) according to the manufacturer’s instructions. Qualified libraries were amplified on cBot to generate clusters on the flowcell (TruSeq PE Cluster Kit V3–cBot–HS, Illumina). The amplified flowcell was then sequenced paired-end on the HiSeq 2000 System (TruSeq SBS KIT-HS V3, Illumina). The entire process was performed by Beijing Genomics Institute (BGI). The RNA-Seq raw data are available on Gene Expression Omnibus (GEO) with accession number GSE123178.

### 2.3. Read Alignment and Transcript Assembly

Fastp (version 0.20.0) was used to filter out low-quality and unclean reads [18]. Hisat2 (version 2.1.0) was used to align the high-quality and clean reads against the reference genome of *C. cinerea*, #326 *Amut Bmut pab1-1*, from JGI (https://genome.jgi-psf.org/Copci_AmutBmut1/Copci_AmutBmut1.home.html, accessed on 30 December 2023) (NCBI Accession PRJNA258994) [3,19]. All transcripts were defined as genes with JGI protein ID and annotated using the Gene Ontology (GO) database to define their roles in biological processes, molecular functions, and cellular components. The genes were also annotated using the EuKaryotic Orthologous Groups (KOG) database to define their roles in metabolism, cellular processes and signalling, and information storage and processing.

### 2.4. Differential Expression Analysis and Functional Gene Annotation

After mapping to the reference genome, the gene expression levels of each sample were calculated by Cufflinks (version 2.2.1) [20]. Differentially expressed genes (DEGs) between the treatment and control samples were then calculated using the Cuffdiff function in Cufflinks [21]. Only transcripts with a *p*-value < 0.05 and |log2 fold change| > 1 were considered statistically significant and used in GO and Kyoto Encyclopedia of Genes and Genome (KEGG) enrichment analyses.

### 2.5. GO and KEGG Enrichment Analysis

GO and KEGG enrichment of DEGs were performed using the R package clusterProfiler (version 3.8.1) [22]. The enrichGo function in clusterProfiler (version 3.8.1) was used to identify the hypergeometric distribution of DEGs based on the org.Cci.sgd.db Bioconductor annotation package (version 2.12.1) [22], whereas the enrichKEGG function was used to calculate the statistical enrichment of DEGs for KEGG pathways. The threshold of *p*-value was set at 0.001 and 0.05 for GO and KEGG enrichment analysis, respectively. The *p.* adjust method of Benjamini and Hochberg (BH) was used to control the false discovery rate [23].

### 2.6. cDNA Synthesis and qRT-PCR Validation

Quantitative real-time PCR (qRT-PCR) of selected genes from the enriched KEGG pathways was used to validate the RNA-Seq data. Approximately 500 ng of total RNA was used to synthesise cDNA using the iScript gDNA Clear cDNA Synthesis Kit (Bio-Rad, Hercules, CA, USA) according to the manufacturer’s instructions. SYBR green-based qRT-PCR was then performed using SsoAdvanced Universal SYBR Green Supermix (Bio-Rad, Hercules, CA, USA) on Applied Biosystems^TM^ 7500 fast Real-Time PCR System (Applied Biosystems, Waltham, MA, USA). Beta-tubulin was used as an endogenous control for normalisation. Information of the primers used is presented in Appendix A.

### 2.7. Statistical Analysis

Data are presented as mean ± SD of three replicates for each treatment. Statistical analysis was carried out using one-way analysis of variance (ANOVA) with *p*-values < 0.05 considered significant.

## 3. Results and Discussion

### 3.1. Effects of LiCl on Mycelium Growth

LiCl in different concentrations (8.5 mM, 17 mM, 35 mM, 70 mM, and 140 mM; water in control) was spread on the surface of YMG plates, on which a block of mycelium from the stock plates was inoculated. After 24 h, no differences in the colony size were observed among the plates (Figure 1). However, after 48 h, the *C. cinerea* mycelium that had been treated with 35 mM or 70 mM LiCl produced larger colonies than those in other concentrations of LiCl and control plates. After 2–3 days, the mycelium that had been treated with 8.5 mM, 17 mM, or 140 mM LiCl produced larger colonies than those of the control. Consecutively, the mycelium that had been treated with 35 or 70 mM LiCl reached the edge of the plates after five days of incubation. However, the mycelium in other plates required six days to reach the edge of the plates. The result suggests that the mycelium in the 35 mM and 70 mM LiCl plates had a higher growth rate than that in other concentrations of LiCl and control plates.

### 3.2. Effects of LiCl on Fruiting Body Initiation

To induce fruiting body formation, plates were transferred to an incubator in a 12/12 h light/dark cycle at 28 °C. After 3–4 days of incubation, initials were observed on the control plates and hyphal knots were observed on plates with 8.5 mM LiCl. In contrast, plates with other concentrations of LiCl were still in the mycelium stage. After 5–6 days, young fruiting bodies were observed on the control plates and the primordium was observed on the 8.5 mM LiCl plates (Figure 2.). Initials and hyphal knots were observed on the 17 mM and 35 mM LiCl plates, respectively. The 70 mM and 140 mM LiCl plates were still in the mycelium stage. Moreover, the mycelial growth in the 140 mM LiCl plates was slightly faster than that in the control plates, and the mycelium stopped growing after reaching the edge of the plates. After one week of incubation, the fruiting bodies in the control plates and plates treated with 8.5 mM, 17 mM, and 35 mM LiCl autolysed, whereas the 70 mM and 140 mM LiCl plates remained in the mycelium stage with no fruiting body formation. Under the same conditions, plates treated with 70 mM KCl formed young fruiting bodies like the control plates did (Figure 1B).

### 3.3. Sequencing of the C. cinerea Transcriptome

To study the transcriptional response of *C. cinerea* to LiCl, RNA from the *C. cinerea* mycelium treated with water and 70 mM LiCl in the mycelium and hyphal knots stages were sequenced with an Illumina HiSeq 2000 sequencer. RNA-Seq resulted in 2.6–3.8 × 10^7^ reads and 3.9–5.8 × 10^9^ bases. Subsequent quality filtering removed reads that were deemed unclear (0.02%). About 94% of the reads were mapped to the reference genome from JGI, resulting in 91,240 transcripts and 14,128 genes. All genes were annotated to GO and KOG, with 2150 functional GO terms and 25 KOG classes.

### 3.4. Differential Expression of Genes in Mycelium under the Influence of LiCl

Of the 14,128 genes expressed in the mycelium treated with water (HM) and 70 mM LiCl (LM1), 1199 were significantly differentially expressed (*p* < 0.05, |log2 fold change| > 1), including 526 upregulated and 673 downregulated genes in LM1 (Appendix A). GO annotation showed that both the upregulated and downregulated DEGs had similar distributions in the molecular function (~70%), biological process (~23%), and cellular component (7%) categories (Appendix A). The DEGs were also annotated into 667 KOG terms, including 261 upregulated and 406 downregulated KOG terms. Almost half of the upregulated DEGs of LM1/HM, that is, expressed higher in LiCl plates, were annotated into metabolism, 22% as cellular processes and signalling, and 14% as information storage and processing (Figure 3A). Downregulated DEGs of LM1/HM had a similar number of genes (about 33%) annotated into metabolism, and cellular processes and signalling, with 9% annotated into information storage and processing (Figure 3B).

### 3.5. GO Functional Enrichment Analysis Explained the Fast Mycelial Growth under the Influence of LiCl by Increased Oxidative Activities

GO functional enrichment analysis showed two categories, biological process and molecular function, with *p*-value ≤ 0.001 (Figure 4). Most of the upregulated DEGs between LM1 and HM (LM1/HM-UP) were enriched in molecular function, including “cofactor binding”, “heme binding”, “tetrapyrrole binding”, “monooxygenase activity”, two “oxidoreductase activity”, and “coenzyme binding”. Downregulated DEGs between LM1 and HM (LM1/HM-DOWN) were enriched in two molecular functions (“carboxy-lyase activity” and “carbon-carbon lyase activity”) and two biological processes (“carbohydrate metabolic process” and “carbohydrate biosynthetic process”) (Figure 4A). No DEGs were significantly enriched in the cellular component category, indicating that the genes of the *C. cinerea* mycelium in this category were not affected at the transcriptional level of by LiCl treatment.

Further analysis of the enriched GO terms revealed that genes related to peroxidase TAP (protein ID: 407604), cytochrome P450 monooxygenase (protein ID: 517014), arylalcohol dehydrogenase (protein ID: 369437), FAD-binding domain-containing protein (protein ID: 369533), oxidoreductase (protein ID: 475201), flavoprotein NADH-dependent oxidoreductase (protein ID: 450208), and flavin-containing monooxygenase (protein ID: 362377) were significantly upregulated (~2-fold increase) in LM1/HM (*p* < 0.05). As the mycelium grows under high temperature, the heat shock causes oxidative damage, affecting growth or even causing cell death in Ganoderma oregonense and Pleurotus species [24]. Likewise, the exposure of *C. cinerea* to an elevated temperature of 37 °C triggers oxidative stress, thereby influencing the growth of its mycelium. Monooxygenase is an enzyme that scavenges reactive oxygen species (ROS) in cells by the concurrent oxidation of NAD(P)H. Arylalcohol dehydrogenase has demonstrated the catalysing ability of the oxidation of alcohols into aldehydes, utilising NAD(P)H as a co-factor that ensures a steady redox system [25,26,27]. The effect of the lithium-induced expression of these genes and the co-expression of these genes might relieve the oxidative stress and result in a faster mycelium growth.

### 3.6. KEGG Pathway Enrichment Analysis Revealed an Association between Metabolism Pathways and Enriched Mycelial Growth under LiCl Influence

KEGG pathway enrichment analysis showed six pathways statistically enriched in the LiCl-treated samples, including the upregulated DEGs in the “biosynthesis of secondary metabolites”, “biosynthesis of antibiotics”, and “tyrosine metabolism”, and the downregulated DEGs in “glycerophospholipid metabolism”, “starch and sucrose metabolism”, and “pyruvate metabolism” (Figure 5). The result indicates that the enhanced mycelial growth of *C. cinerea* treated with 70 mM LiCl was mainly associated with metabolism pathways, especially those in secondary metabolism.

KEGG enrichment analysis showed that the enriched DEGs in LM1 were concentrated in metabolism pathways. Two enzymes were upregulated, and six were downregulated in the “starch and sucrose metabolism pathway” (Appendix A). UDP-glucose is the substrate of glycogen synthase in the glycogen synthesis process [28]. The downregulated expression of glycogen synthase (protein ID: 457685) detected here may then allow more UDP-glucose to be used in the synthesis of beta (1,3)-D-glucan, a component of the cell wall, for mycelium growth [29]. In addition, the upregulation of glucoamylase (protein ID: 473268) and downregulation of alpha-amylase (protein ID: 495249) allow the breakdown of glycogen into glucose for mycelium growth instead of dextrin. Moreover, the upregulation of beta-glucosidase (protein ID: 373418) breaks down beta-D-glucosides or cellobiose into glucose, thereby allowing rapid cell division and cell mass accumulation [30].

Galectin-1 (protein ID: 473274) and galectin-2 (protein ID: 488611) (Appendix A) were upregulated in the LiCl-treated samples under sufficient nutrients and darkness. Galectin is considered a developmental gene that regulates cell growth and extracellular cell adhesion [31]. Previous studies suggest that the expression of galectin genes is differentially regulated by environmental signals and is upregulated under nutrient depletion and light when hyphal aggregates to form a primordium of *C. cinerea* [32,33]. However, the expression of galectin-1 (protein ID: 473274) and galectin-2 (protein ID: 488611) in LM1 was 5.8- and 5.7-fold higher than that in the control, respectively, implying that the effect of LiCl mimics the environmental signal to induce the expression of galectin even in the mycelium under excess nutrients and darkness. The results support that galectins are not sufficient for fruiting body formation.

### 3.7. Differential Expression of Genes in Fruiting Body Initiation under the Influence of LiCl

A total of 1391 genes were significantly differentially expressed between hyphal knots of the water-treated samples (HHK) and the consistent mycelium of LiCl-treated samples (LM2) (*p* < 0.05, |log2 fold change| > 1), of which 658 were upregulated and 733 were downregulated in LM2 (Appendix A).

GO annotation showed that both the LiCl upregulated and downregulated DEGs had a similar distribution in molecular function (71–75%), biological process (17~22%), and cellular component (7~8%) (Appendix A). DEGs at the two time points were compared by using the Venn diagram. There were 208 upregulated and 102 downregulated DEGs in the LM1 and LM2 samples compared with the control at the two different time points, respectively (Figure 3). A total of 20 common DEGs in LM1/HM-down and LM2/HHK-UP changed from downregulated to upregulated, indicating that these 20 genes may be essential for *C. cinerea* to maintain their vegetative hypha form. In contrast, 59 common DEGs in LM1/HM-UP and LM2/HHK-DOWN changed from upregulated to downregulated, indicating that these 59 genes may be essential for *C. cinerea* fruiting body initiation and LiCl interrupted their expression.

The DEGs were annotated into 769 KOG terms, 355 of which were upregulated and 414 were downregulated in LM2/HHK. Forty-one percent of the LM2/HHK-UP DEGs belonged to metabolism, followed by cellular processes and signalling (25%), and information storage and processing (13%) (Figure 3). For the downregulated DEGs, 34% were involved in cellular processes and signalling, 28% in metabolism, and 26% in information storage and processing. The number of upregulated DEGs in LM2/HHK that were annotated into KOG classes was lower than that in LM1/HM-UP. However, the number of downregulated DEGs in LM2/HHK-DOWN that were annotated into KOG classes was higher than that in LM1/HM-DOWN (Figure 6 and Figure 7).

### 3.8. GO Functional Enrichment Analysis Revealed Most Downregulated Biological Processes in Fruiting Body Initiation in LiCl-Treated Samples

Further detailed GO functional enrichment analysis of the upregulated DEGs between LM2 and HHK, LM2/HHK-UP, showed 10 statistically enriched GO terms with *p*-value ≤ 0.001, including “flavin adenine dinucleotide binding”, “coenzyme binding”, and “oxidoreductase activity, acting on CH-OH group of donors” from the molecular function category and “mitochondrial respiratory chain complex III assembly” from the biological process category (Figure 4B). For the downregulated DEGs between LM2 and HHK, LM2/HHK-DOWN, 11 enriched GO terms were from the biological process category, including “DNA metabolic process”, “carbohydrate metabolic process”, “cell cycle checkpoint”, and “negative regulation of cell cycle”, and three enriched GO terms were from the cellular component category, namely “cell wall”, “external encapsulating structure”, and “fungal-type cell wall” (Figure 4B). The enriched terms indicated that LiCl caused the consistent mycelium LM2 to better survive oxidative stress and, at the same time, diminished the structural developments needed for fruiting body initiation.

Other than these terms specifically enriched at one time point by LiCl treatment, one GO term with downregulated DEGs, LM1/HM-DOWN and LM2/HHK-DOWN, from the biological process category “carbohydrate metabolic process”, and six GO terms with upregulated DEGs from the molecular function category were enriched at both time points, LM1/HM-UP and LM2/HHK-UP—“cofactor binding”, “heme binding”, “tetrapyrrole binding”, “iron ion binding”, “monooxygenase activity”, and “oxidoreductase activity” changed in expression (Figure 4), suggesting that carbohydrate metabolism would need to continuously decrease and oxidative stress responses would need to continuously increase in fruiting body initiation.

### 3.9. KEGG Pathway Enrichment Analysis Revealed an Association between Cellular Process Pathways and Inhibited Fruiting Body Initiation under LiCl Influence

KEGG pathway enrichment analysis of DEGs between LM2 and HHK resulted in six statistically enriched pathways in the LiCl-treated samples (Figure 5). The upregulated DEGs, LM2/HHK-UP, were DEGs in “Steroid biosynthesis” from the metabolism pathways category. Five other pathways enriched in LM2/HHK-DOWN included “Cell cycle-yeast”, “DNA replication”, “Meiosis-yeast”, “Mismatch repair”, and “Homologous recombination” from the cellular process category (Appendix A), indicating that LiCl suppressed the expression of these DEGs involved in the cell cycle and meiosis which are essential for fruiting body initiation.

### 3.10. Downregulated DEGs Were Associated with the Inhibition of Fruiting Body Development

The effects of LiCl on fruiting body initiation could also be shown with expression analysis of individual genes. The amount of time required for fruiting body formation increased when the concentration of LiCl increased. Upon 70 mM LiCl, *C. cinerea* fruiting body formation was inhibited even under favourable environmental conditions. When the mycelium transits to hyphal knots, it requires the expression of genes for the construction of new membranes. A previous transcriptome study has shown that genes that are involved in phospholipid biosynthesis are upregulated in fruiting body initiation of *C. cinerea* [3]. In this study, genes of the phospholipid biosynthesis process, including phosphatidylserine decarboxylase (protein ID: 361342, 457157, 178286, and 178424), phosphatide cytidylyltransferase (protein ID: 442684), inositol-3-phosphate synthase (protein ID: 479579), and a hypothetical protein (protein ID: 178424) (Appendix A), were downregulated in LM2 treated with 70 mM LiCl compared with the control samples that formed hyphal knots, LM2/HHK-DOWN. Most of these genes participate in glycerophospholipid metabolism. Therefore, we suggest that LiCl affects glycerophospholipid metabolism, disrupting the transition from the mycelium to hyphal knots. In addition, fruiting body induction requires important chromosome remodelling genes, such as *Cc.arp9* and *Cc.snf5* [2,34,35]. These genes were also downregulated in this study (Appendix A), indicating that LiCl may be associated with the expression of chromosome remodelling genes for cell division, which results in the formation of hyphal knots [36].

### 3.11. LiCl Inhibited Signal Transduction Responding to Environmental Factors

Environmental factors, such as light, temperature, and nutrients, individually or in combination, influence fruiting body formation in basidiomycetes. In this study, *C. cinerea* was first grown at 37 °C under total darkness with rich nutrients and then at 28 °C in a light/dark cycle after the mycelium reached the edge of plates. Light, cold shock, and nutrient-diminishing increase the expression of stress response genes and trigger or promote fruiting [1,33,37,38]. The expressions of stress response genes are upregulated by certain environmental conditions during fruiting body formation, including nutrients and light [1,3,4]. Nutrients in the substrate are linked to G-protein-coupled receptor, G protein alpha, adenylyl cyclase, cyclic AMP (cAMP), cAMP-dependent protein kinase, and Ras, which are involved in the cAMP-PKA signalling pathway [39]. The homolog Gα proteins Gpa1 and Gps3 of *Cryptococcus neoformans* and *Ustilago maydis*, respectively, belong to a preserved subgroup of fungal “cAMP-type” Gα proteins that is involved in responding primarily to nutrient deficiency by increasing cAMP levels and activating cAMP-dependent protein kinase (PKA) [40]. Previous physiological studies have reported that the intracellular cAMP levels increase in *C. cinerea* during hyphal knot and primordium formation in response to glucose, under the induction by activated Gα protein [1,39]. However, recent studies in *Schizophyllum commune* and *C. cinerea* have revealed the opposite action of PKA that the downregulation of PKA suppresses its downstream targets during fruiting body development [4,41]. Our data showed that the expression of nutrient response genes such as PKA protein kinase (protein ID: 461884) was significantly downregulated and adenylate cyclase (protein ID: 497554) was slightly downregulated in LM2/HHK-DOWN (Appendix A). The expression of other components in the cMAP-PAK pathway in LM2 samples was similar to HHK. We hypothesised that LiCl inhibits PKA, and in turn its downstream targets, from transferring signals to transcriptional factors that control the transcription of hyphal knot formation genes, and interrupts hyphal knot formation in LM2 samples even under nutrient deprivation.

Other than nutrition, light is a crucial environmental signal that triggers fruiting body development. Many research efforts have contributed to the identification of fungal light reception in *Neurospora crassa* [42]. The light receptor complex, White collar-1/White collar-2 (WC-1/WC-2), regulates spore formation, photoadaptation, and the circadian clock [42,43,44,45]. For basidiomycetes, two light response genes, *dst1* and *dst2* in *C. cinerea*, encode the homologue of WC-1 and flavin adenine dinucleotide (FAD)-binding-4-domain, respectively [46,47]. A homologue of WC2 has been characterised in *C. cinerea*. Disruption of this homologue leads to abnormal fruiting body development [34,48]. In addition, Yang et al. showed that during the switch from mycelial growth to fruiting body development, *CmWC-1*, a homologue of the blue-light receptor gene in *Cordyceps militaris*, suppresses the expression of genes related to steroid biosynthesis [49]. Under light, the light receptor WC-1 homologue in *Trichoderma atroviride* induces a cellular response through WC-dependent blue-light signalling, involving the HOG and MAPK pathways, and regulates the phosphorylation of HOG MAP kinase Tmk3 and MAP kinase [50]. Our results showed that the expression of HOG MAP kinase homologue (protein ID: 449305) and two MAP kinase homologues (protein ID: 429744) in *C. cinerea* was slightly upregulated in LM2/HHK-UP, implying that the light receptor in LM2 activated the Hog and MAPK pathways to transfer light signalling. However, DEGs of steroid biosynthesis were upregulated in LM2 samples, indicating that the downstream target of HOG and MAPK pathways may fail to transmit the light signal to repress the expression of steroid-biosynthesis-related genes under the influence of LiCl, so *C. cinerea* remained in the mycelium stage even though the blue light receptor protein receives the signal from light.

### 3.12. Kinases Were Downregulated in Mycelium and Fruiting Body Initiation

The heat map in Figure 8 shows the expression of protein kinases in the LiCl-treated samples and control sample at two different time points (Figure 8). Seven different protein kinase DEGs were upregulated in LM1 and LM2. However, many more protein kinases were downregulated in LM1 and LM2. Among the LM1 samples, 12 protein kinase DEGs were downregulated, including CAMK1 protein kinase and PASK protein kinase in the CAMK group, STE7 protein kinase in the STE group, MAPK protein kinase in the CMGC group, a protein kinase in the conventional protein kinase (ePK) group, HisK protein kinase and Mak3 protein kinase in the HisK group, ABC1-C protein kinase and inositol pentakisphosphate 2 kinase in the PIKK group, as well as phosphoenolpyruvate carboxykinase and acetate kinase in the atypical protein kinase group (aPKs). Among LM2 samples, the downregulated protein kinase DEGs included a PKA protein kinase in the AGC group, a MARK protein kinase in the CAMK group, two SRPK protein kinases in the CMGC group, a TKL-ccin protein kinase in the TKL group, and nine protein kinases in the ePK group.

Some groups of protein kinases have been identified for their roles in the cellular processes. The CMGC group protein kinases, such as CDK, MAPK, GSK3, and CLK, have functions in cell cycle control and MAPK signalling splicing [51]. LiCl reduced the demand of SRPK protein kinases in the CMGC group in the cell cycle. The expression of cell cycle and meiosis genes was downregulated in the LM2 samples. In addition, protein kinases in AGC and CAMK groups, such as PKA and MAPK protein kinases, take part in signalling pathways well characterised in intracellular signalling, transducing the signal from the cell surface to nucleus [52]. Over several decades, comprehensive research efforts have been made to explain the mechanisms underlying the fruiting body development of *C. cinerea*. Recent large-scale functional genetic analyses have provided insights into the overall biological structure of *C. cinerea* [6]. However, a complete picture of *C. cinerea*’s fruiting body developmental networks remains elusive, primarily because the functions of kinases, which play a central role in signalling and metabolic pathways, are not described on a genome-wide scale. Here, by studying the effects of LiCl on the fruiting body formation in *C. cinerea* accompanied with extensive kinase expression changes, we began to show that protein kinases may be crucial to fruiting body initiation.

### 3.13. The Effect of LiCl Revealed That the Activity of GSK3 Could Be an Important Channel in Fruiting Body Development

Lithium chloride is a well-known chemical used in mammalian studies [53,54]. Apart from the effect of lithium on the Glycogen synthase kinase 3 (GSK3), lithium has also been suggested to inhibit myo-inositol monophosphatase (IMPase) [55,56]. Lithium was used to study the regulation of GSK3 and IMPase in the fungi *Schizophyllum commune*, *Fusarium graminearum*, and *Magnaporthe oryzae*, and showed that lithium reduced mycelial growth [41,57,58]. However, in our study, the mycelium treated with lithium grew faster than the control, which is different from the other studies. In *Schizophyllum commune*, a conserved signature, histidine acid phosphatase (HAP), is involved in the myo-inositol phosphatase signalling [59]. The repression of HAP was considered correlated with the increase in the higher phosphorylation of inositol phosphates in the LiCl-treated *Schizophyllum commune* [59]. However, in our study, the expression of a hypothetical protein (protein ID: 447513), which was annotated into histidine acid phosphatase, was slightly differentially expressed in LM1 samples and was similar in both LM2 and HHK samples. The validation of the gene expression profile in *C. cinerea* transcriptome data by qRT-PCR showed similar results as RNA-Seq data (Appendix A). The expressions of myo-inositol monophosphatase (protein ID: 397251) and inositol pentakisphosphate-2-kinase (protein ID: 461885) were downregulated in LM1 samples when compared with HM samples, implying that the inhibition of IMPase by LiCl promoted the mycelial growth of *C. cinerea*. During the fruiting body formation stage, the LM2 samples treated with LiCl remained as mycelium indefinitely. Williams et al. [60] pointed out that the lithium effect on target inhibition was stage-determinant. Lithium inhibited IMPase during early development processes; however, GSK3 inhibition by lithium affected later development. In our study, the gene expressions of myo-inositol monophosphatase (protein ID: 397251) and inositol pentakisphosphate-2-kinase (protein ID: 461885) in the LM2 and HHK samples were similar, indicating that the effect of lithium on IMPase may occur in the early vegetative stage of *C. cinerea*, and then the effect of lithium may switch to GSK3 in the later development stage of the fruiting body. In previous studies, the expression of GSK3 (protein ID: 362899) in *C. cinerea* was conservatively regulated [13]. The function of GSK3 is in the regulation of growth, conidiogenesis, mycotoxin production, pathogenicity, and the stress response mechanism in phytopathogenic fungi *Fusarium graminearum* and *Magnaporthe oryzae*, and GSK3 is considered a control regulator in cellular processes [57,58]. For example, in *Fusarium graminearum*, the signal GSK3 homolog is suggested to promote the transcription of numerous early-meiosis-specific genes as the *Δfgk3* mutant fails in sexual development processes [57,61]. Moreover, genes encoded as glycogen synthase kinase in ascomycetous fungi, such as *Saccharomyces cerevisiae* and *Neurospora crassa*, regulate protein–protein interaction, meiotic gene activation, and transcription factors of the circadian clock under environmental signals like nutrient, temperature, and light [62,63,64]. Because of the well-known effect of LiCl on GSK3 activities, our results of the effect of LiCl on fruiting body formation, and the studies about the role of GSK3 in fungal development, we propose that the activity of GSK3 may be an important link between extracellular signals and respective gene expressions related to fruiting body formation.

## 4. Conclusions

To the best of our knowledge, this study represents the first transcriptome profiling of the effects of LiCl on *C. cinerea* mycelial growth and fruiting body formation. We found enriched metabolism pathways, such as “biosynthesis of secondary metabolite” and “biosynthesis of antibiotic”, in which DEGs were upregulated with enhanced mycelial growth by the addition of LiCl at the mycelium stage. Moreover, the DEGs that were involved in cellular processes, including “cell cycle-yeast” and “meiosis-yeast”, were suppressed by LiCl, and may be the reason that *C. cinerea* remained in the mycelium stage even under favourable environmental conditions for fruiting body formation. Enrichment analyses showed that LiCl inhibited glycogen formation, channelling the carbon to glucose which supported faster mycelial growth. Our data showed many inhibition targets of LiCl related to fruiting body formation, including G proteins, kinases, stress-responding genes, and cell division genes. The large number of transcript changes caused by LiCl can be caused by its action on multiple targets but can also be caused by its action on a few targets that started cascades of changes. We would like to propose that the well-known predominant LiCl inhibitory target, GSK3, could be a key protein kinase for the transduction of environmental signals to affect the expression of fruiting body initiation genes and genes in fruiting-related cellular processes. Further studies on the phosphorylation of GSK3 responding to external environmental signals are needed to better understand the mechanism of environmental factors inducing fruiting body formation.

## Figures and Tables

**Figure 1 jof-10-00140-f001:**
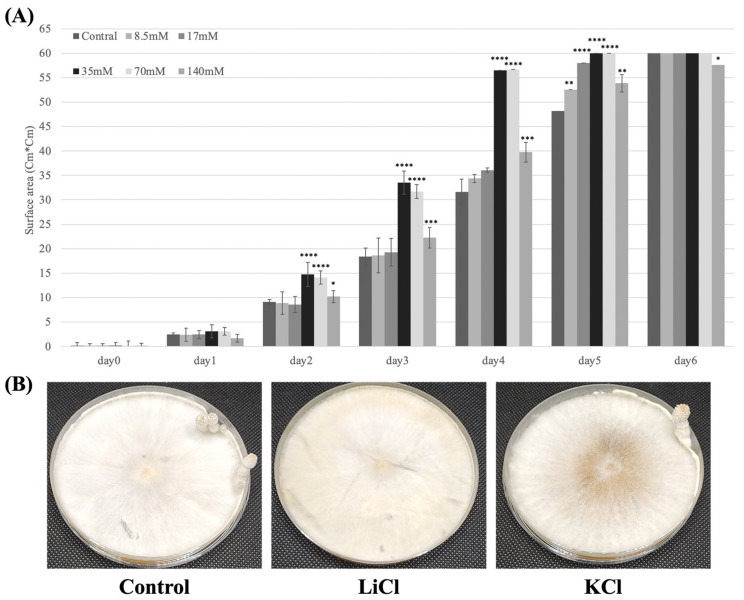
(**A**) Mycelium radial growth rate of *C. cinerea* in LiCl-treated samples and control; Control: treated with water; the concentration of LiCl: 8.5 mM, 17 mM, 35 mM, 70 mM, and 140 mM. Vertical bars represent the standard deviation of mean (*n* = 3, *: *p* < 0.05, **: *p* < 0.01, ***: *p* < 0.001, ****: *p* < 0.0001, by one-way ANOVA). (**B**) Young fruiting bodies were observed on the YMG plate treated with 70 mM KCl and the control YMG plates were treated with water; YMG plates treated with 70 mM LiCl remained in the mycelium stage.

**Figure 2 jof-10-00140-f002:**
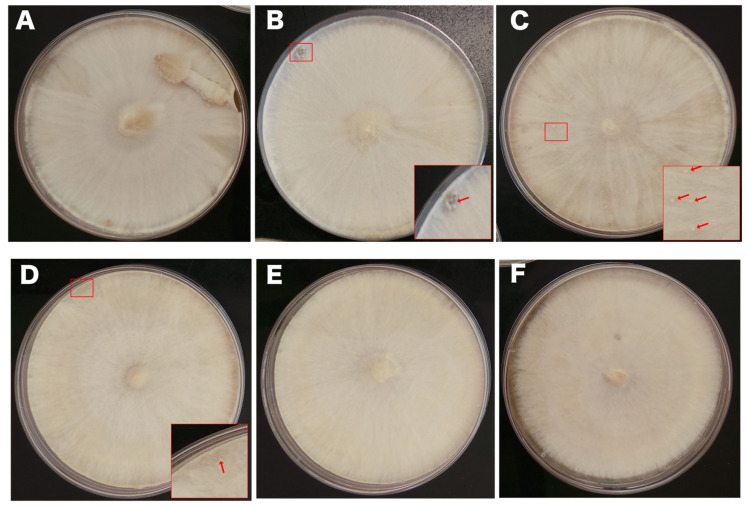
Effect of different concentrations of LiCl on *C. cinerea* fruiting body development. (**A**) Young fruiting bodies were observed on the control YMG plates treated with water. (**B**) Primordium was observed on the YMG plates treated with 8.5 mM LiCl after five days of incubation; red arrow indicates primordium. (**C**,**D**) Initials and hyphal knots were observed on the YMG plates treated with 17 mM and 35 mM LiCl, respectively; red arrow indicates initials and hyphal knots. (**E**,**F**) YMG plates treated with 70 mM and 140 mM LiCl remained in the mycelium stage, and mycelium treated with 140 mM LiCl stopped reaching the edge of plates.

**Figure 3 jof-10-00140-f003:**
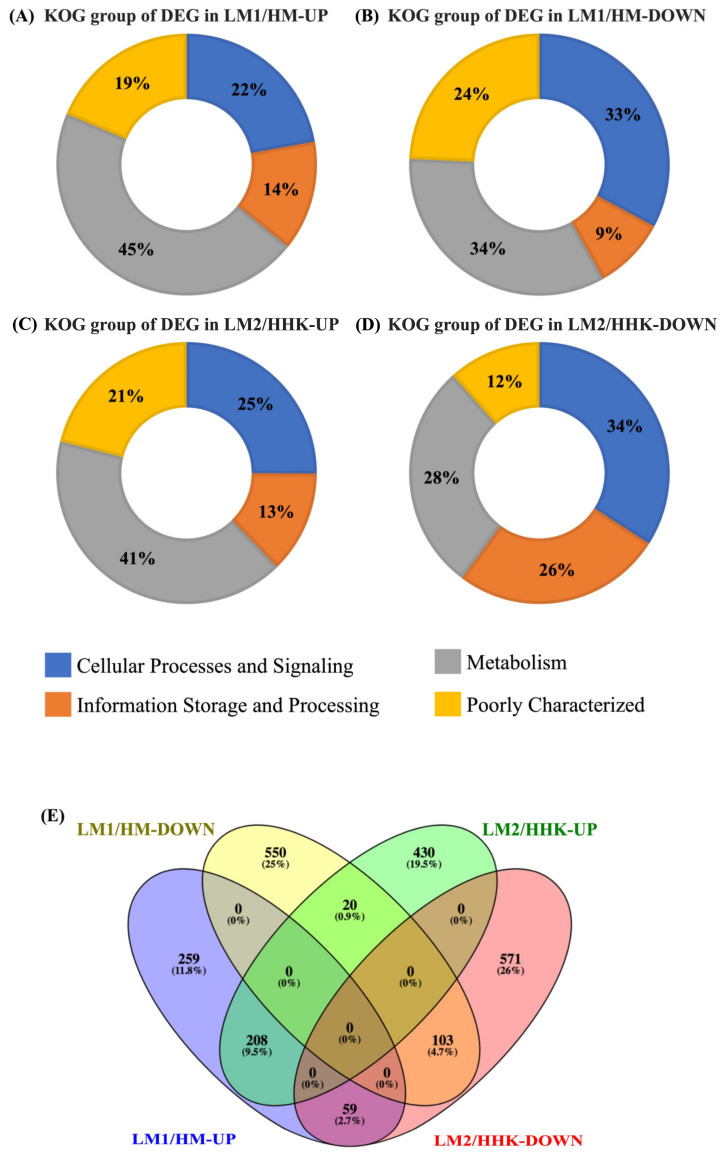
KOG annotation of DEGs. The distribution of KOG annotations of (**A**) upregulated and (**B**) downregulated DEGs in LM1 vs. HM, HM: *C. cinerea* mycelium treated with water, LM1: *C. cinerea* mycelium treated with 70 mM LiCl; and (**C**) upregulated and (**D**) downregulated DEGs in LM2 vs. HHK, HHK: hyphal knots of water-treated samples, LM2: consistent mycelium of 70 mM LiCl-treated samples. (**E**) Venn diagram illustrating overlapping of upregulated and downregulated DEGs at two time points.

**Figure 4 jof-10-00140-f004:**
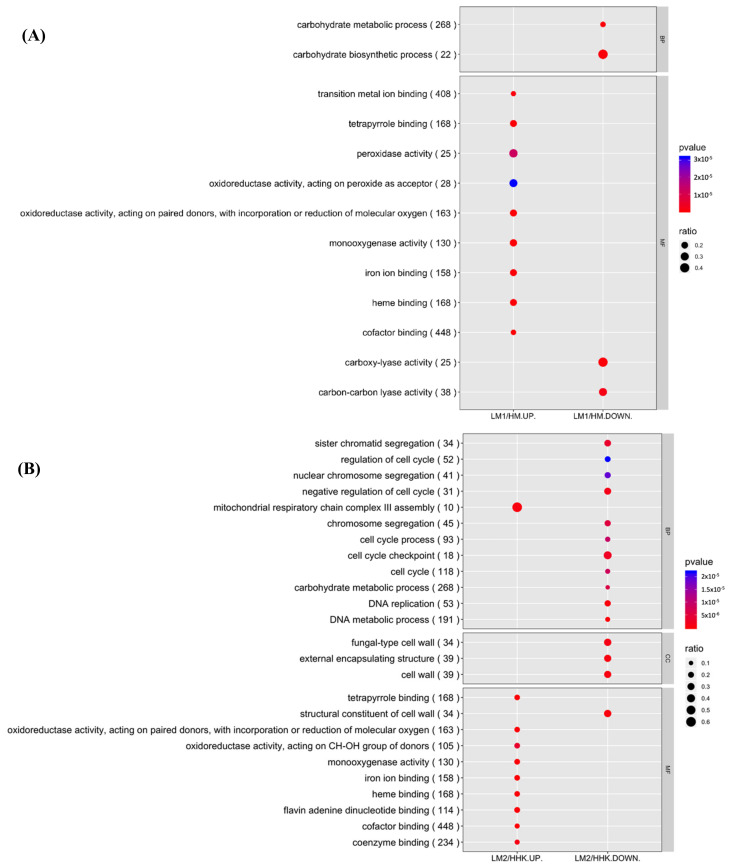
GO functional enrichment analysis. The size of the dots indicates the ratio of DEGs involved in the GO term, and the colour represents the *p*-value. (**A**) LM1/HM. HM: *C. cinerea* mycelium treated with water; LM1: *C. cinerea* mycelium treated with 70 mM LiCl. (**B**) LM2/HHK. HHK: hyphal knots of water-treated samples; LM2: consistent mycelium of 70 mM LiCl-treated samples. UP: upregulated DEGs, higher expression in the presence of LiCl; DOWN: downregulated DEGs, lower expression in the presence of LiCl.

**Figure 5 jof-10-00140-f005:**
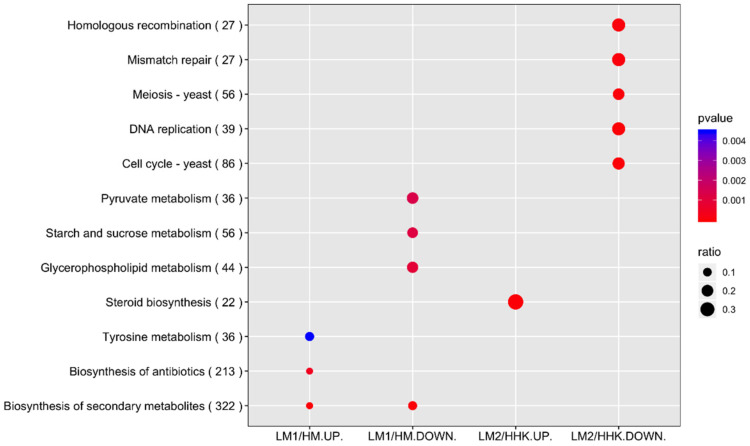
KEGG pathway enrichment analysis. The size of the dots indicates the number of DEGs involved in the pathway, and the colour represents the *p*-value. HM: *C. cinerea* mycelium treated with water; LM1: *C. cinerea* mycelium treated with 70 mM LiCl; HHK: hyphal knots of water-treated samples; LM2: consistent mycelium of 70 mM LiCl-treated samples; UP: upregulated DEGs; DOWN: downregulated DEGs.

**Figure 6 jof-10-00140-f006:**
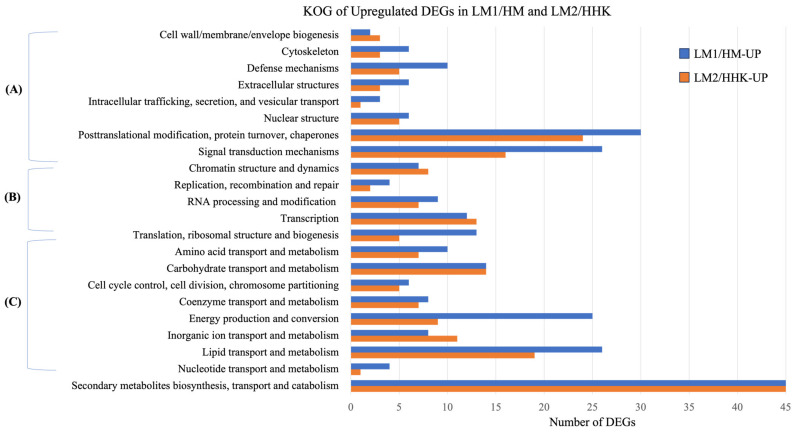
KOG classification of upregulated DEGs in LiCl-treated samples LM1/HM and LM2/HHK. (**A**) Cellular processes and signalling; (**B**) information storage and processing; (**C**) metabolism. HM: *C. cinerea* mycelium treated with water; LM1: *C. cinerea* mycelium treated with 70 mM LiCl; HHK: hyphal knots of water-treated samples; LM2: consistent mycelium of 70 mM LiCl-treated samples; UP: upregulated DEGs.

**Figure 7 jof-10-00140-f007:**
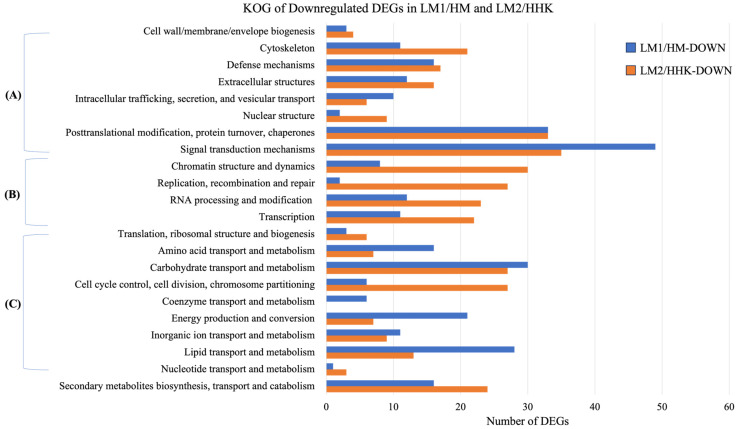
KOG classification of downregulated DEGs in LiCl-treated samples, LM1/HM and LM2/HHK. (**A**) Cellular processes and signalling; (**B**) information storage and processing; (**C**) metabolism; HM: *C. cinerea* mycelium treated with water; LM1: *C. cinerea* mycelium treated with 70 mM LiCl; HHK: hyphal knots of water-treated samples; LM2: consistent mycelium of 70 mM LiCl-treated samples; DOWN: downregulated DEGs.

**Figure 8 jof-10-00140-f008:**
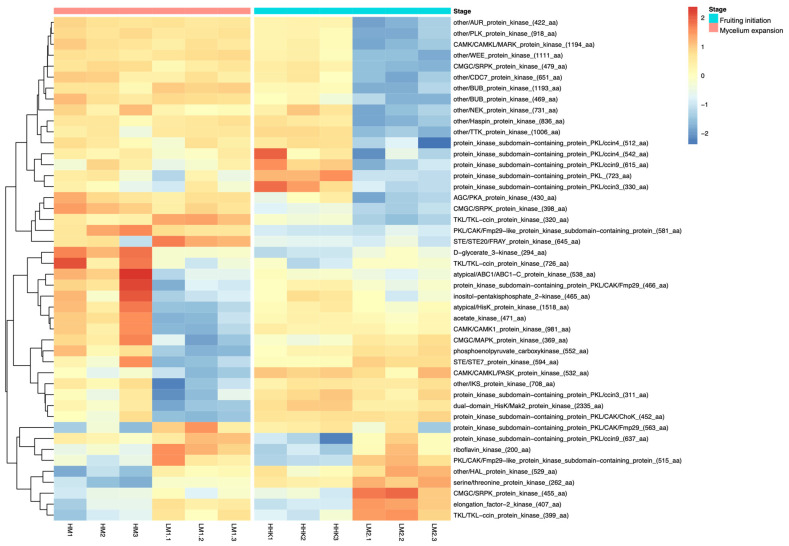
Heat map showing the expression pattern of protein kinase DEGs involved in each sample. HM: *C. cinerea* mycelium treated with water; LM1: *C. cinerea* mycelium treated with 70 mM LiCl; HHK: hyphal knots of water-treated samples; LM2: consistent mycelium of 70 mM LiCl-treated samples. Red represents high expression; Blue represents low expression.

## Data Availability

Sequencing data from this study have been submitted to the NCBI Gene Expression Omnibus (GEO, https://www.ncbi.nlm.nih.gov/geo/, accessed on 30 December 2023) with accession number GSE123178.

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
