# Peer review of "Transcriptome Analysis Reveals Mycelial and Fruiting Responses to Lithium Chloride in Coprinopsis cinerea"

_jof, 2024, doi:10.3390/jof10020140_

Round 1
Reviewer 1 Report
Comments and Suggestions for Authors
Specific comments:
Line 62 and 67 – What concentration of LiCl?
Line 110– Please add citation for medium.
Line 71 – “In this study we found….” – this sentence should be moved to results section. The last paragraph of introduction should inform what authors intended to find out.
Line 84– What medium was used? It is not clear. It seems as it was composed only of LiCl.
Line 91 and 100– RNA was isolated only from mycelium or from mycelium and fruiting bodies? It’s important.
Line 119 – Which genome? Accession number?
Fig 1A – Bars are hidden in upper edge.
Fig 1B – There is culture variant containing KCl, but it is not mentioned in material and methods section. Moreover, it seems that it also affected fungal growth.
Fig 3 – For me it has low resolution.
Line 259 – “….significantly…” – how many times?
Line 262 – “When…” – whole sentence should be rewritten. It is not clear.
Line 264 – Aryl alcohol dehydrogenase is an enzyme catalyzing oxidation of aromatic alcohols to corresponding aldehydes using NAD. It’s primary role is to oxidize alcohol not to produce NADH. The citation for this sentence concerns different enzyme.
Author Response
Thank you for your comments. I have revised the manuscript according to your comments.
Line 62 and 67 – What concentration of LiCl?
Response: Apart from mushroom-forming fungi, LiCl has also been reported to inhibit the growth of Trichoderma species, however, several wood-decaying basidiomycetes species can tolerate LiCl at 6g/L.
Response: Our previous study has reported that a particular concentration of LiCl at 3g/L inhibits fruiting body formation in C. cinerea
Line 110– Please add citation for medium.
Response: RNA sequencing libraries were constructed by using the TruSeq RNA Sample Prep Kit v2 (Illumina, Inc., USA) according to the manufacturer’s instructions.
Line 71 – “In this study we found….” – this sentence should be moved to results section. The last paragraph of introduction should inform what authors intended to find out.
Response: In this study, we investigated the impact of LiCl promoted on mycelial growth and fruiting body initiation in C. cinerea. We used RNA-Seq to examine the effects of LiCl on the transcriptional responses of C. cinerea in mycelial growth, and fruiting body formation.
Line 84– What medium was used? It is not clear. It seems as it was composed only of LiCl.
Response: LiCl with different concentrations was dissolved in autoclaved water
Line 91 and 100– RNA was isolated only from mycelium or from mycelium and fruiting bodies? It’s important.
Response: RNA was extracted from mycelium at first time point and hyphal knot at second time point respectively.
Line 119 – Which genome? Accession number?
Response: Hisat2 was used to align the high quality and clean reads against the reference genome of C. cinerea, #326 Amut Bmut pab1-1, from JGI (https://genome.jgi-psf.org/Copci_AmutBmut1/ Copci_AmutBmut1.home.html) (NCBI Accession PRJNA258994)
Fig 1A – Bars are hidden in upper edge.
Response: updated in manuscript.
Fig 1B – There is culture variant containing KCl, but it is not mentioned in material and methods section. Moreover, it seems that it also affected fungal growth.
Response: “Concentration of potassium chloride (KCl) up to 70 mM also prepared for comparison.” Added into material and methods section.
Fig 3 – For me it has low resolution.
Response: updated on manuscript
Line 259 – “….significantly…” – how many times?
Response: Further analysis of the enriched GO terms revealed that genes related to peroxidase TAP (protein ID: 407604), cytochrome P450 monooxygenase (protein ID: 517014), arylalcohol dehydrogenase (protein ID: 369437), FAD-binding domain-containing protein (protein ID: 369533), oxidoreductase (protein ID: 475201), flavoprotein NADH-dependent oxidoreductase (protein ID: 450208), and flavin-containing monooxygenase (protein ID: 362377) were significantly upregulated (~2-fold increase) in LM1/HM (P< 0.05).
Line 262 – “When…” – whole sentence should be rewritten. It is not clear.
Response: Exposure of C. cinerea to an elevated temperature of 37°C triggers oxidative stress, thereby influencing the growth of its mycelium.
Line 264 – Aryl alcohol dehydrogenase is an enzyme catalyzing oxidation of aromatic alcohols to corresponding aldehydes using NAD. It’s primary role is to oxidize alcohol not to produce NADH. The citation for this sentence concerns different enzyme.
Response: Arylalcohol dehydrogenase have demonstrated the catalyzing ability to oxidation of alcohols into aldehydes, utilizing NAD(P)H as a co-factor that ensures steady redox system [48, 69, 70].
Reviewer 2 Report
Comments and Suggestions for Authors
The reviewed manuscript, titled “Transcriptome analysis reveals mycelial and fruiting responses to lithium chloride in Coprinopsis cinerea” by Chan et al. is a fundamentally sound study on the transcriptional analysis of the response of this fungus to LiCl exposure. Overall, this work is well conceived, designed, performed, and analyzed. The author’s writing style is clear and concise throughout the paper, and they present their findings and analysis in a straightforward manner that is easy to read. There are a few minor grammatical and formatting errors, and I have highlighted a few of them below. There is nothing egregious, however I would encourage the authors to review their revised work with ‘fresh eyes’ – either themselves or a colleague – to address these errors. They detract a bit from the manuscript and should be resolved.
Overall the strengths of the paper are the growth assays presented in figure #1, the fruiting body development in figure #2, and the transcriptional analysis performed by RNA-Seq. The figures and analysis of the RNA-Seq are straightforward and fundamentally sound, however the writing in this section tends to come across a bit repetitive and is not as concise as the work elsewhere. I would encourage the authors to critically read and revise this section. Some of the verbiage can simply be integrated into the figures or into a table and referenced. This would allow the authors to easily highlight their findings in a central easy to access area and focus the text onto the analysis – which is one of the strong points of this manuscript!
These comments are relatively minor and should be easy to accomplish. Aside from that here are a few comments from the reading of the work. Comments:
Figure 2: line 198. Panels B, C, and D contain red arrows that seem to be highlighting something, yet they are not discussed in the figure legend. Please add an explanation to the legend to indicate the significance of these markers.
Line 208-209: I believe that there are typos when indicating the number of reads that were generated from the RNA-Seq experiments. It looks like the # of reads should be 2.6-3.8x10^7 rather than 2.6-3.8x107, this is a crucial distinction. Please revise for accuracy on this and the next line.
Line 271: The DEGs in the “biosynthesis of secondary metabolites” is interesting and relevant to this reviewer’s work. Oftentimes these genes are clustered to allow/facilitate proper stochiometric levels of expression for several reasons, e.g. reduction of toxic intermediates, limiting cellular stressors, energy efficiency and conservation. As a result of this, I would encourage the authors to consider a brief discussion. I would suggest to look at a specific pathway or two for their genomic localization to see if there are any biosynthetic clusters in this – or any of the other DEG grouping families. No further analysis needs to be performed, however I think that it would be quite beneficial to address, this reviewer has several publications in the field that may be of interest, however I would like to emphasize that these are suggestions – the authors are under no obligation to use or cite either of these and my decision will not be affected by their inclusion or exclusion. These are simply being suggested as my work directly relates to this field:
1. Cittadino, Gina M., et al. "Functional Clustering of Metabolically Related Genes Is Conserved across Dikarya." Journal of Fungi 9.5 (2023): 523.
2. Asfare, Sarah, et al. "Systematic analysis of functionally related gene clusters in the opportunistic pathogen, Candida albicans." Microorganisms 9.2 (2021): 276.
Line 454: I believe there is a typo after LM2. Please double check formatting
Line 466: The heatmap should be enlarged as a figure to allow the readers to more clearly see and read this analysis. In this reviewer’s opinion, this is one of the more significant components of the manuscript and it deserves more space for clarity.
Line 485: italicize C. cinera.
Comments on the Quality of English LanguageEnglish needs a thorough review, there are relatively minor inconsistencies, however they are consistent throughout the work and should be resolved.
Author Response
Thank you for your time and constructive feedback. We have revised the manuscript according to your comments.
These comments are relatively minor and should be easy to accomplish. Aside from that here are a few comments from the reading of the work. Comments:
Figure 2: line 198. Panels B, C, and D contain red arrows that seem to be highlighting something, yet they are not discussed in the figure legend. Please add an explanation to the legend to indicate the significance of these markers.
Response: Revised in manuscript “Figure 2. Effect of different concentrations of LiCl on C. cinerea fruiting body development. (A): Young fruiting bodies were observed on the control YMG plates treated with water. (B): Pri-mordium was observed on the YMG plates treated with 8.5 mM LiCl after five days of incuba-tion; red arrow indicated primordium. (C) and (D): Initials and hyphal knots were observed on the YMG plates treated with 17 mM and 35 mM LiCl, respectively; red arrow indicated initials and hyphal knots. (E) and (F): YMG plates treated with 70 mM and 140 mM LiCl remained in the mycelium stage, and mycelium treated with 140 mM LiCl stopped reaching the edge of plates.”
Line 208-209: I believe that there are typos when indicating the number of reads that were generated from the RNA-Seq experiments. It looks like the # of reads should be 2.6-3.8x10^7 rather than 2.6-3.8x107, this is a crucial distinction. Please revise for accuracy on this and the next line.
Response: Sorry for typo and revised in manuscript
Line 271: The DEGs in the “biosynthesis of secondary metabolites” is interesting and relevant to this reviewer’s work. Oftentimes these genes are clustered to allow/facilitate proper stochiometric levels of expression for several reasons, e.g. reduction of toxic intermediates, limiting cellular stressors, energy efficiency and conservation. As a result of this, I would encourage the authors to consider a brief discussion. I would suggest to look at a specific pathway or two for their genomic localization to see if there are any biosynthetic clusters in this – or any of the other DEG grouping families. No further analysis needs to be performed, however I think that it would be quite beneficial to address, this reviewer has several publications in the field that may be of interest, however I would like to emphasize that these are suggestions – the authors are under no obligation to use or cite either of these and my decision will not be affected by their inclusion or exclusion. These are simply being suggested as my work directly relates to this field:
- Cittadino, Gina M., et al. "Functional Clustering of Metabolically Related Genes Is Conserved across Dikarya." Journal of Fungi9.5 (2023): 523.
- Asfare, Sarah, et al. "Systematic analysis of functionally related gene clusters in the opportunistic pathogen, Candida albicans." Microorganisms9.2 (2021): 276.
Response: Thank you for your great suggestions regarding the "biosynthesis of secondary metabolites" and the potential clustering of DEGs. We are glad to know your interest in our work and the valuable perspective you bring to the discussion. However, we have carefully considered our research focus and currently believe that delving into this specific aspect may divert our attention from the objectives of our study. we have decided not to include a detailed discussion on this particular aspect in our current manuscript.
Line 454: I believe there is a typo after LM2. Please double check formatting
Response: Revised in manuscript
Line 466: The heatmap should be enlarged as a figure to allow the readers to more clearly see and read this analysis. In this reviewer’s opinion, this is one of the more significant components of the manuscript and it deserves more space for clarity.
Rsponse: Revised in manuscript
Line 485: italicize C. cinera.
Response: Revised in manuscript